# Structure-based design of a strain transcending AMA1-RON2L malaria vaccine

Palak N. Patel [1], Thayne H. Dickey [1], Ababacar Diouf[2], Nichole D. Salinas[1], Holly McAleese [3], Tarik Ouahes[3], Carole A. Long [2], Kazutoyo Miura [2], Lynn E. Lambert[3] & Niraj H. Tolia [1] ✉

Apical membrane antigen 1 (AMA1) is a key malaria vaccine candidate and target of neutralizing antibodies. AMA1 binds to a loop in rhoptry neck protein 2 (RON2L) to form the moving junction during parasite invasion of host cells, and this complex is conserved among apicomplexan parasites. AMA1-RON2L complex immunization achieves higher growth inhibitory activity than AMA1 alone and protects mice against *Plasmodium yoelii* challenge. Here, three single-component AMA1-RON2L immunogens were designed that retain the structure of the two-component AMA1-RON2L complex: one structure-based design (SBD1) and two insertion fusions. All immunogens elicited high antibody titers with potent growth inhibitory activity, yet these antibodies did not block RON2L binding to AMA1. The SBD1 immunogen induced significantly more potent strain-transcending neutralizing antibody responses against diverse strains of *Plasmodium falciparum* than AMA1 or AMA1-RON2L complex vaccination. This indicates that SBD1 directs neutralizing antibody responses to strain-transcending epitopes in AMA1 that are independent of RON2L binding. This work underscores the importance of neutralization mechanisms that are distinct from RON2 blockade. The stable single-component SBD1 immunogen elicits potent strain-transcending protection that may drive the development of next-generation vaccines for improved malaria and apicomplexan parasite control.

*Plasmodium falciparum* malaria remains one of the deadliest and most prevalent infectious diseases globally[1]. The risk of contracting malaria and developing severe illness is considerably higher for infants, children, and pregnant women[1]. In addition to the increased risk for these populations, the emergence of antimalarial drug resistance undermines malaria control efforts around the world[1]. This emphasizes the need for an effective vaccine that prevents parasites from establishing infection or progressing to the invasion of red blood cells and protects against clinical malaria.

Adults living in malaria-endemic areas develop robust immunity against clinical disease over the course of multiple natural infections[2-4]. A vaccine that induces a similar immune response could successfully prevent malaria pathogenesis. Furthermore, the transfer of purified immunoglobulin G (IgG) from malaria-immune adults to nonimmune individuals with acute blood stage malaria greatly reduced parasitemia and clinical symptoms[5-7]. This indicates that merozoite surface antigens are prime targets of protective antibody responses in blood-stage malaria immunity. The malaria merozoite

[1]Host-Pathogen Interactions and Structural Vaccinology Section, Laboratory of Malaria Immunology and Vaccinology, National Institute of Allergy and Infectious Diseases, National Institutes of Health, Bethesda, MD, USA. [2]Laboratory of Malaria and Vector Research, National Institute of Allergy and Infectious Diseases, National Institutes of Health, Rockville, MD, USA. [3]Vaccine Development Unit, Laboratory of Malaria Immunology and Vaccinology, National Institute of Allergy and Infectious Diseases, National Institutes of Health, Bethesda, MD, USA. ✉e-mail: niraj.tolia@nih.gov

protein apical membrane antigen 1 (AMA1) is critical for RBC invasion and is one of the most promising blood-stage vaccine candidates[8,9]. AMA1 has been extensively studied for its role in red cell invasion[10–15] and a role for AMA1 in sporozoite infection of the liver and for transmission to mosquitoes has recently been reported[16,17]. AMA1 and its role in invasion are conserved among apicomplexan parasites that cause diverse diseases of human and agricultural relevance[18–20]. This suggests that AMA1-based vaccines have the potential to elicit multistage protection against natural malaria parasite infection and clinical malaria and against diverse apicomplexan parasites.

Malaria and apicomplexan parasites invade target host cells using a moving junction (MJ) formed between the apex of the parasite and the host cell membrane[18–24]. MJ is initiated by the export of the rhoptry neck proteins RON2, RON4, and RON5 into the host cell. RON2 spans the host cell membrane and serves as a receptor for AMA1, which is membrane anchored on the surface of the parasite[12,21–24]. AMA1 binds to RON2 through a surface exposed loop (RON2L) to anchor the parasite to the host cell membrane prior to internalization into a parasitophorous vacuole[21–24]. The AMA1 ectodomain structure has a stacked three-domain architecture comprised of three disulfide-constrained domains (domains I-III)[12,25,26]. AMA1 undergoes extensive proteolytic processing in the merozoite, including removal of the N-terminal prosequence[8,27–30]. Domains I (DI) and II (DII) of AMA1 form a RON2L binding site that is partially occupied by the DII loop that extends from domain II[15,31]. The DII loop is highly flexible and undergoes conformational changes to expose the binding site for RON2[12,15,26,31]. Antibodies or peptides that prevent the formation of the AMA1-RON2 complex block red cell invasion by parasites[32–36]. Antibodies against AMA1 are also believed to block red cell invasion by disrupting secondary proteolytic processing on the merozoite surface[37].

The presence of AMA1 on the merozoite surface and the ability of AMA1-specific antibodies to neutralize parasites in vitro and in vivo indicate that AMA1 is a potential vaccine candidate[8,38–40]. An AMA1-based vaccine FMP2.1/AS02$_A$[41] elicited strong and sustained antibody responses in naïve individuals[42,43] and in malaria-exposed adults and children[44–46]. However, AMA1 alleles in endemic areas are highly polymorphic. These polymorphisms serve as an immune evasion strategy to circumvent strain-transcending protection, preventing the development of effective strain-transcending vaccines based on AMA1[47–50]. Antibody responses elicited by single AMA1 alleles show significantly lower efficacy against heterologous strains[47]. To address this problem and achieve strain-transcending protection, combinations of up to seven AMA1 alleles or the design of three diversity covering (DiCo) variants to elicit strain-transcending antibody responses were evaluated with limited success[33,48,51–55]. Malaria antigens also display the immune evasion phenomenon of antigenic diversion, where the action of neutralizing antibodies is prevented by interfering non-neutralizing antibodies that enable parasite survival[56]. Careful design of antigens based on AMA1 and RON2 to account for these immune evasion mechanisms may result in strain-transcending protection.

AMA1-based vaccines induce strong antibody responses but do not provide significant protection against clinical malaria in controlled infection studies, and their efficacy in field studies is lower than expected[43,46,57,58]. Variations in the dose, adjuvant, and formulation of AMA1-based vaccines showed only moderate improvements[48,59–61]. In contrast, rats immunized with the two-component AMA1-RON2L complex elicited higher levels of anti-AMA1 neutralizing antibodies than those immunized with AMA1 alone, likely because the AMA1-RON2L complex better mimics the true AMA1 structure on invading merozoites[62]. Additionally, mice immunized with a *Plasmodium yoelii* AMA1-RON2L complex show complete antibody-dependent protection against a lethal *Plasmodium yoelii* challenge[62]. Furthermore, immunizing *Aotus* monkeys with the AMA1-RON2L complex protects against a virulent *Plasmodium falciparum* infection and shows higher

neutralizing activity in vitro than AMA1 alone[63]. These studies suggest that enhancement of the quality of the antibody response toward greater neutralizing antibodies may be required over simply improving the quantity of the antibody response.

Here, we created single-component immunogens that mimic the AMA1 complex structure on the invading merozoite. Three independent designs were evaluated: one structure-based design (SBD1) of AMA1 to reconfigure the sequence permitting attachment of RON2L to the C-terminus and two insertion fusions placing RON2L within the sequence of AMA1. These single-component AMA1-RON2L immunogens possess improved characteristics over AMA1 and replicate the structure of the two-component AMA1-RON2L complex to varying extents. The RON2L in all designed immunogens occupies the binding site in an irreversible manner, making the designed immunogens incapable of binding exogenous RON2 peptides and immunoglobulin new antigen receptor (IgNAR) 14I-1, which both engage the open binding site in AMA1. We examined the antibody quantity and quality elicited by these immunogens in rats. The designed immunogens do not elicit antibodies that block RON2L binding to AMA1, consistent with locked RON2 bound in the fused immunogens. Despite the lack of RON2L blocking activity, the antibodies raised against the single-component immunogens provided protective GIA with *Plasmodium falciparum 3D7* similar to AMA1 DI-DII and the AMA1 DI-DII-RON2L complex. Strikingly, the SBD1 immunogen showed significantly more potent strain-transcending GIA with heterologous *Plasmodium falciparum FVO* and *Dd2* parasites than either the AMA1 DI-DII or AMA1 DI-DII-RON2L complexes. These results demonstrate that antibodies targeting regions of AMA1 DI-DII outside of the RON2 binding site and DII loop contribute substantially to strain-transcending and cross-neutralizing activity. These single-component immunogens form the basis for the next generation of AMA1-based antigens for strain-transcending protection against malaria and other apicomplexan parasites.

## Results

### Design of single-component immunogens with improved biophysical characteristics by combining RON2L with AMA1 DI-DII

We created three single-component immunogens (Fig. 1) containing domains I and II (DI-DII) of AMA1 fused to RON2L. The RON2L binding site in apo-AMA1 comprises a domain I hydrophobic groove and a region that is exposed when the DII loop (Lys351 to Ala387) is displaced by RON2L. Upon displacement, the DII loop adopts a disordered state, does not contact RON2L and appears dispensable for binding. In the absence of RON2L, the DII loop is stabilized by domain I[64,65]. RON2L contacts discontinuous residues in AMA1 that are located in the middle of the protein sequence.

A single-component AMA1-RON2L immunogen cannot be created by simple fusion of RON2L to the N- or C-terminus of AMA1 because the AMA1 termini are located far from the RON2L binding site and would require a large linker to facilitate the correct orientation of RON2L in the pocket. We used structure-based design (SBD) to alter the location of the C-terminus, enabling seamless attachment of RON2L. The SBD1 immunogen is a circular permutation of AMA1 that contains a Gly/Ser linker (GGGGS × 4) between the original N- and C-termini. The DII loop (358-TDYEKIKEGFKNKNASMIKSAFLPTGAF-385) is removed in SBD1 to produce novel N- and C-termini at residues Lys386 and Thr357, respectively. This new AMA1 C-terminus is immediately adjacent to the N-terminal helix of bound RON2L. Some of the residues deleted in SBD1 (360-YEKIKEGFK-368) comprise a helix in AMA1, which is replaced by the N-terminal helix of RON2L (4-QQAK-DIGAG-12). This design approach ensures that the RON2L sequence (3-TQQAKDIGAGPVASCFTTRMSPPQQICLNSVVNTALS-39) could be appended without a linker to create SBD1 (Fig. 1)[15].

Additionally, we created insertion fusion immunogens by inserting RON2L into an AMA1 loop proximal to the RON2L binding site.

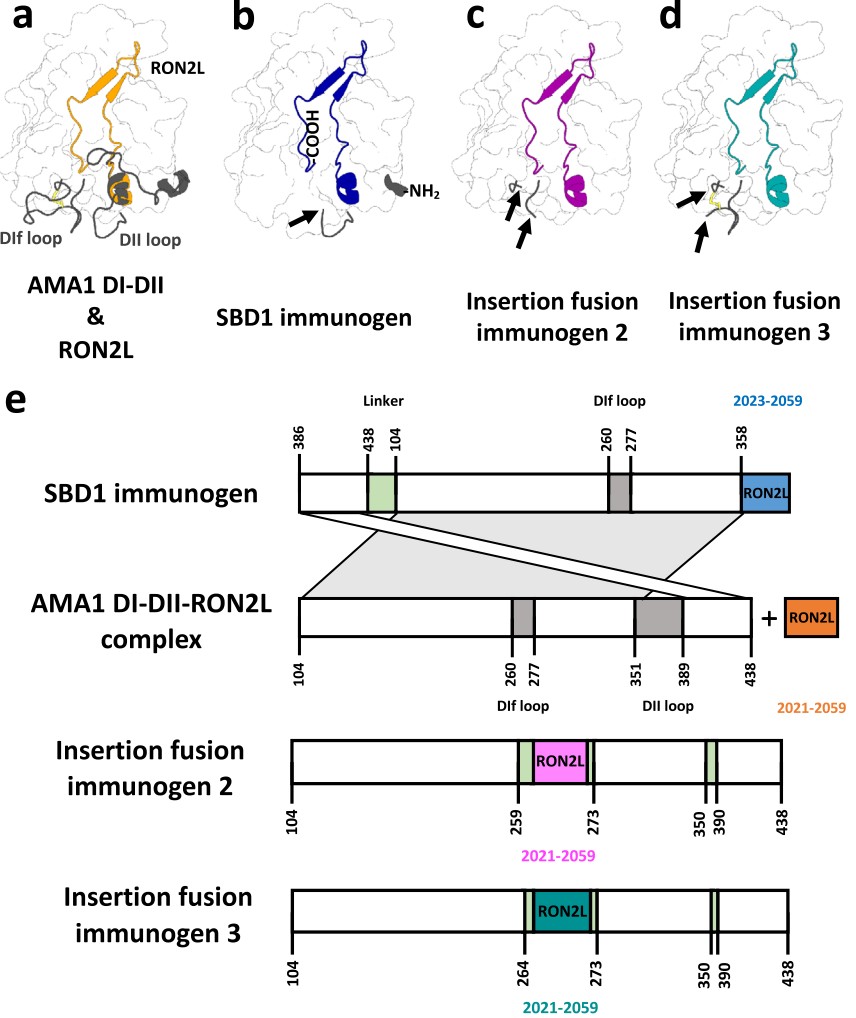

Fig. 1 | **Overview of the design of single-component immunogens. a** Structure of apo AMA1 DI-DII showing the domain II loop (DII loop) and DIf loop (gray cartoon) and the location of RON2L in the bound complex (orange cartoon). **b** A circularly permuted SBD1 immunogen was created by introducing a Gly/Ser linker between the original termini (not shown) and by removing the DII loop, which produced novel N- and C-termini at residues Lys386 and Thr357, respectively. Then, RON2L was fused to this new C-terminus without a linker. **c, d** Insertion fusion immunogens 2 and 3 were constructed by replacing the DIf loop of AMA1 DI-DII with RON2L and

by removing the DII loop. Insertion fusion immunogen 3 retains Cys263 and its disulfide bridge (yellow). This figure was created using structures of apo AMA1 (PDB ID: 4r19) and the AMA1-RON2L complex (PDB ID: 3zwz). An arrow indicates the point of fusion. **e** Schematic illustrating the design processes for the three immunogens discussed in this article. Proteins are shown from the N- to C-terminus, and numbers indicate residue numbering based on the wild-type AMA1 and RON2 sequences.

Insertion fusion immunogens 2 and 3 were constructed by replacing several amino acids in the DIf loop of AMA1 with RON2L and a flanking Gly/Ser linker. Immunogen 2 lacks amino acids 260-PRYCNKDESKRNS-272 of the DIf loop, including Cys263, consequently disrupting a disulfide bridge. Immunogen 3 lacks only amino acids 265-KDESKRNS-272, retaining Cys263 and the disulfide bridge. The disordered DII loop was replaced with a Gly/Ser linker in both of these insertion fusion immunogens to prevent the potential displacement of the fused RON2L. We also created an AMA1 DI-DII design in which the DII loop was replaced by a short Gly-Ser linker (AMA1 DI-DII ΔDII-loop) to examine the impact of the removal of the DII loop.

AMA1 DI-DII, AMA1 DI-DII ΔDII-loop and each of these three immunogens (Supplementary Table 1) were expressed in HEK293 cells and purified to homogeneity. The expressed AMA1 DI-DII, AMA1 DI-DII ΔDII-loop and immunogens were folded, monomeric and monodisperse, as evidenced by size exclusion chromatography and SDS-PAGE analysis (Fig. 2a, Supplementary Fig. 1a). All three designed immunogens had a higher mean purification yield than WT AMA1 DI-DII (8.6 mg/l), with SBD1 and insertion fusion immunogens 2 and 3

demonstrating purification yields of 14.2 mg/l, 21.9 mg/l and 16.9 mg/l, respectively (Fig. 2b). All immunogens showed marked improvement in their average melting temperature ($T_m$) by approximately 21 °C, from 52 °C to 74 °C (Fig. 2c, d). The improvement in $T_m$ is primarily a result of the fusion of RON2L and not due to the removal of the DII loop (Supplementary Fig. 1b, c). In summary, fusion of RON2L to AMA1 DI-DII produced three distinct single-component immunogens with substantially improved biophysical characteristics.

**The fused RON2L is bound to AMA1 in the immunogens**

The fusions of RON2L to AMA1 were designed to replicate the bound state of the complex. The bound state is expected to be unable to bind exogenous RON2L and unable to bind antibodies that compete with RON2L binding. The neutralizing immunoglobulin new antigen receptor (IgNAR) 14I-1[34] binds to an epitope in AMA1 located within the hydrophobic RON2L binding groove and competes with RON2L binding. We determined the accessibility of the RON2L binding site in the designed immunogens by probing with IgNAR 14I-1 and exogenous RON2L using biolayer interferometry (BLI) and enzyme-linked

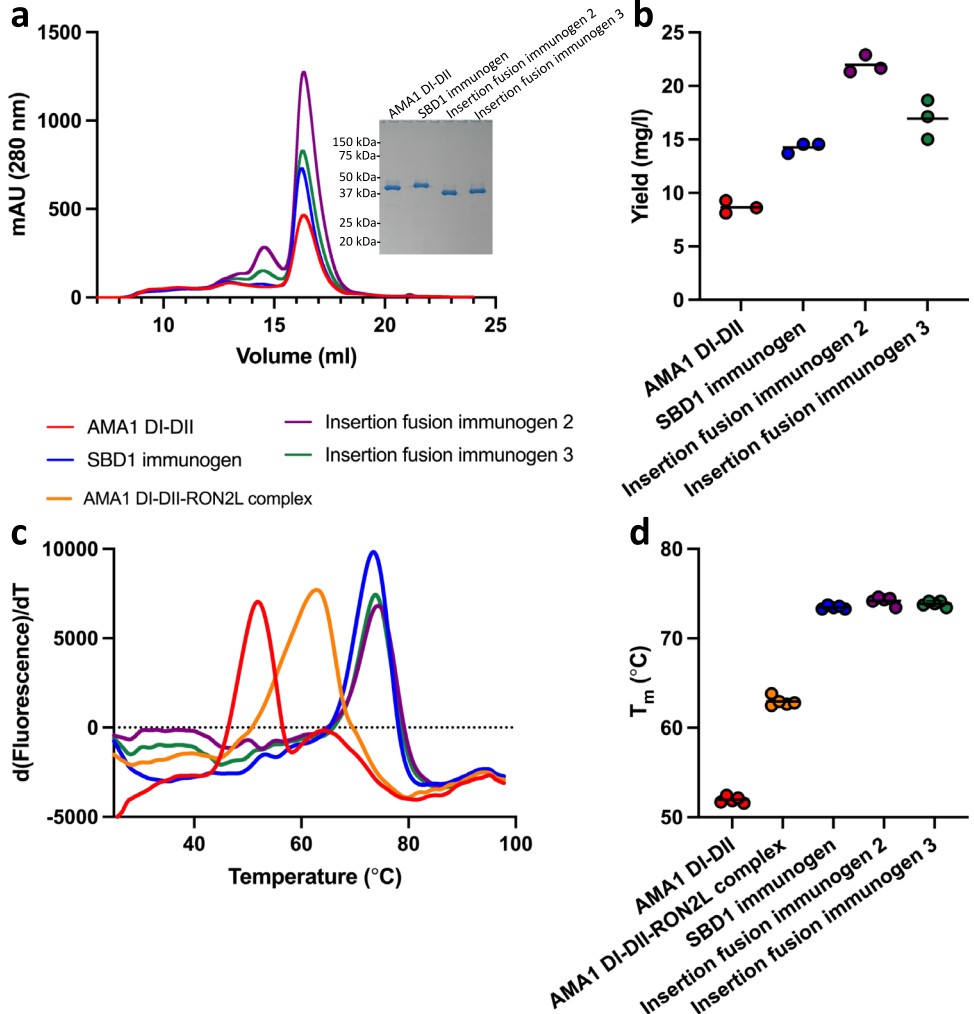

**Fig. 2 | The yield and stability of single-component immunogens are higher than those of AMA1 DI-DII alone and the AMA1 DI-DII-RON2L complex. a** All three immunogens were expressed at higher levels than AMA1 DI-DII and eluted as monomers by size exclusion chromatography (SEC). The inset in (**a**) confirms the high purity of the immunogens through reducing SDS-polyacrylamide gel electrophoresis (PAGE). **b** Purification yield from three separate purifications. Bars represent the mean yield from three separate purifications. **c** Differential scanning fluorimetry indicated that three immunogens have higher thermostability than AMA1 DI-DII and the AMA1 DI-DII-RON2L complex. **d** $T_m$ from five independent measurements. Bars represent the mean. Source data are provided as a Source data file.

immunosorbent assay (ELISA). AMA1 DI-DII, which has an accessible RON2L binding site, was able to effectively bind to 14I-1 with binding clearly observable by BLI at concentrations as low as ~10 nM (Fig. 3a). In contrast, none of the immunogens bound to 14I-1 even at 200 nM, the highest concentration tested (Fig. 3a), demonstrating that the fused RON2L occupied the binding site and prevented accessibility. Similar results were obtained by ELISA, where AMA1 DI-DII bound to 14I-1, while all three immunogens showed little or no binding with 200 nM or 1000 nM 14I-1 (Fig. 3b). This suggests that antibodies with epitopes in the domain I hydrophobic groove are unable to engage the designed immunogens. In a similar manner, AMA1 DI-DII bound to exogenous RON2L by both BLI and ELISA, while the immunogens exhibited little to no binding (Fig. 3c, d). These results indicate that the designs were successful in replicating the bound state of the complex.

**Structures of the designed immunogens recapitulate the AMA1-RON2L complex and reveal the molecular basis for enhanced stability**

We investigated whether fused RON2L is correctly bound to AMA1 in the designed immunogens through structural analysis. We determined the X-ray crystal structures of SBD1 and insertion fusion immunogens 2 and 3 to resolutions of 1.80 Å, 1.85 Å, and 2.10 Å, respectively (Fig. 4a, Table 1). We superimposed the structures of these immunogens on the previously characterized AMA1 DI-DII-RON2L complex (PDB ID: 3zwz)[15]. The overall structures of the designed immunogens were very similar to the native AMA1 DI-DII-RON2L complex. The SBD1 structure was most similar to the AMA1-RON2L complex, with no major structural reorganization observed and a Cα root mean square deviation (RMSD) of 0.299 over 245 C-alpha residues (Fig. 4b, Supplementary Fig. 2a). In contrast, insertion fusion immunogens 2 and 3 retained the RON2L binding mode of the complex but displayed local distortions in loops near the vicinity of the insertion sites, resulting in Cα RMSDs of 0.381 over 232 C-alpha residues and 0.309 Å over 228 C-alpha residues, respectively (Fig. 4b, Supplementary Fig. 2b, c). In all cases, the N-terminal helix of fused RON2L was located at one end of the binding site with the coil extending into a disulfide-closed loop resulting in a U-shaped structure (Fig. 4c, Supplementary Fig. 3). A vast majority of the interface residues between fused RON2L and AMA1 DI-DII in the diverse designs were identical to those found in the AMA1 DI-DII-RON2L complex, indicating that fused RON2L binds correctly to AMA1 DI-DII (Supplementary Tables 2, 3, and 4). All three structures revealed that the two cysteine residues in the RON2L peptide that are necessary

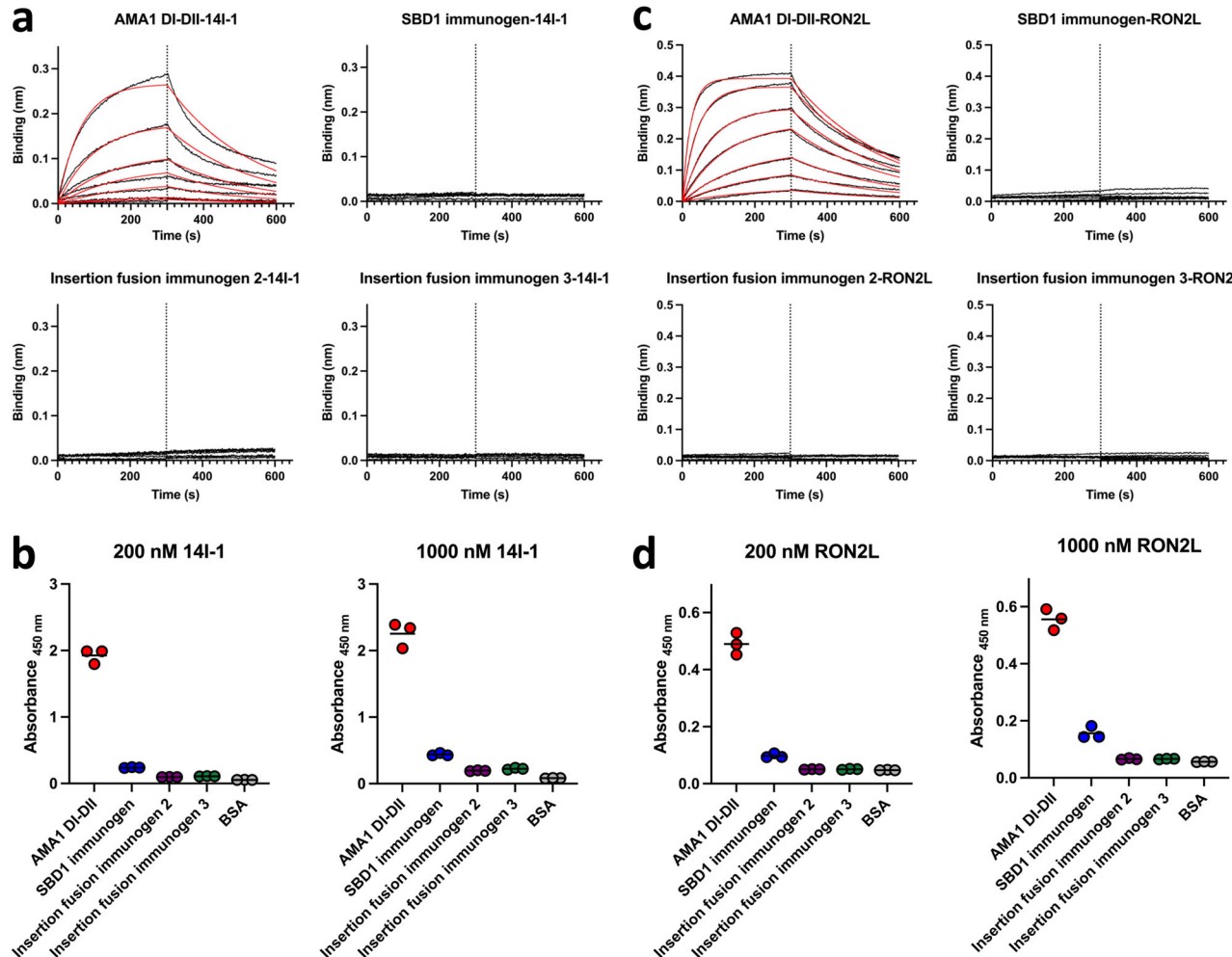

**Fig. 3 | RON2L is bound to AMA1 in the designed immunogens, preventing accessibility to the RON2L binding site. a** Representative biolayer interferometry (BLI) traces used to quantitatively measure the binding of immunogens to IgNAR 14I-1 demonstrating inaccessibility of the epitope located in the RON2L binding pocket in the immunogens. Immunogens were two-fold serially diluted in the range of 200 nM to 3.125 nM. **b** IgNAR 14I-1 shows little or no binding to immunogens in three independent ELISAs. **c** Representative BLI traces used to measure the binding

of immunogens to exogenous RON2L demonstrating that the binding site for exogenous RON2L is occupied by the fused RON2L in the designed immunogens. Immunogens were two-fold serially diluted in the range of 200 nM to 3.125 nM. **d** Exogenous RON2L does not bind to immunogens in three independent ELISAs. In **b** and **d**, bovine serum albumin (BSA) was used as a negative control. Source data are provided as a Source data file.

for binding to AMA1 are disulfide-linked (Supplementary Fig. 3). Additionally, a key interacting Arg residue, corresponding to ARG2041 in RON2, in the fused RON2L of immunogens fits well into a pocket in a manner identical to that in the complex structure (PDB ID: 3zwz) (Supplementary Fig. 4). The binding of RON2L appears to improve residue packing in domain I and enhance the conformational stability of all three structures.

### The designed immunogens produce similar antibody titers to the control groups, indicating that the quantity of the antibody response is unchanged

We examined how these improved biophysical characteristics and altered epitope availability impacted immunogenicity and growth inhibitory activity (GIA). Groups of nine rats were immunized three times at three-week intervals with 20 μg of single-component SBD1 immunogen, insertion fusion immunogens 2 or 3, apo AMA1 DI-DII, or the AMA1 DI-DII-RON2L two-component complex. All antigens were adjuvanted with AddaS03™, which is a research grade mimic of AS03, an adjuvant approved for human use (Fig. 5a). There was no significant difference between the levels of AMA1 DI-DII-specific antibodies induced by the immunogens and the levels induced by the AMA1 DI-DII

or AMA1 DI-DII-RON2L two-component complex. This similarity in titers is noteworthy because the immunogens do not elicit antibodies to the deleted DII loop or the blocked hydrophobic pocket (*vide infra*). These results suggest that the majority of antibodies induced by AMA1 DI-DII target epitopes distinct from the DII loop and RON2L binding site (Fig. 5b).

### Antibodies raised by the designed immunogens do not block RON2L binding by AMA1, indicating a drastically different quality of the antibody response

We measured inhibition of the direct protein–protein interaction between AMA1 DI-DII and RON2L in a blocking assay that measures RON2L binding to AMA1. Blocking antibody titers were determined by serially diluting sera from individual rats after the third vaccination on day 63. Rats immunized with either AMA1 DI-DII or the AMA1 DI-DII-RON2L two-component complex elicited high titers of RON2L blocking antibodies. In contrast, all three immunogens elicited significantly lower levels of blocking antibodies, typically at the limit of detection of the assay, than the AMA1 DI-DII or AMA1 DI-DII-RON2L two-component complex (Fig. 5c). This result indicates that the immunogens do not elicit antibodies that recognize the domain I hydrophobic groove and

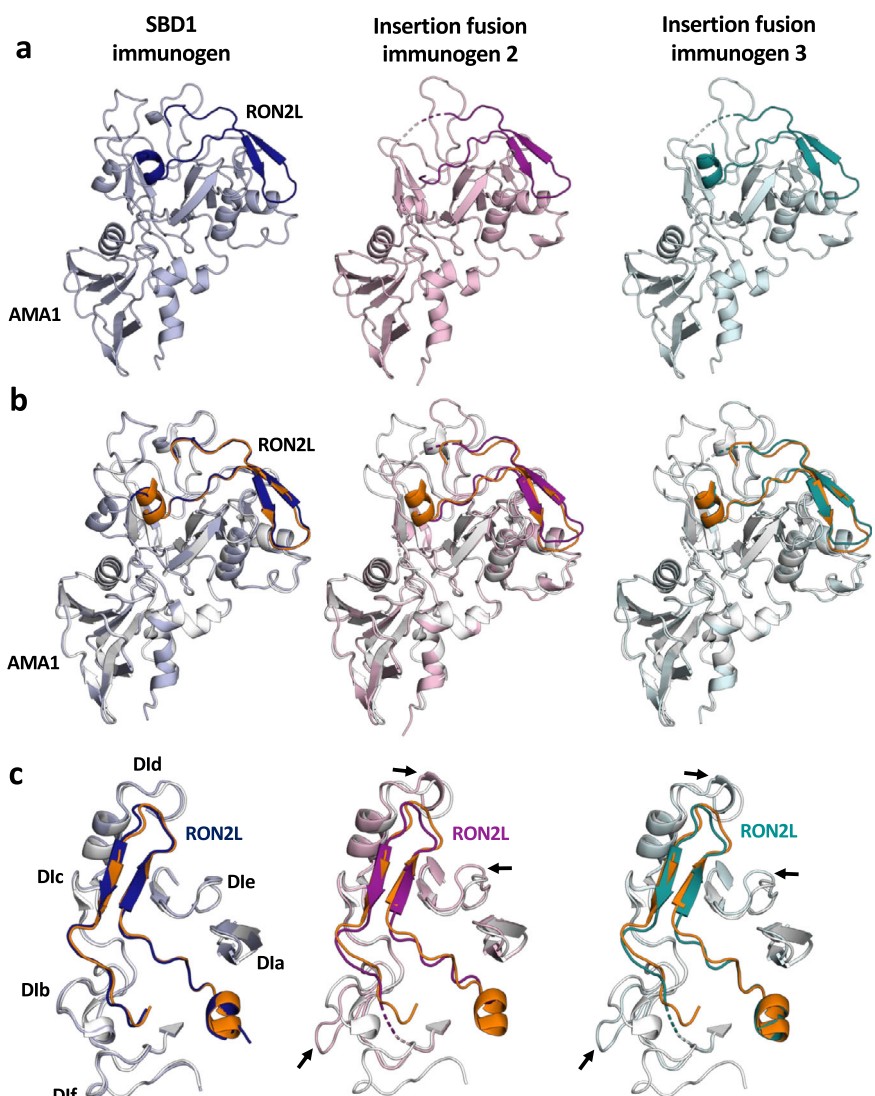

**Fig. 4 | Single-component immunogens have a very similar structure to the AMA1-RON2L complex, with the SBD1 immunogen possessing the greatest structural similarity. a** Crystal structures of the single-component SBD1 immunogen (light blue, blue) and insertion fusion immunogens 2 (light pink, magenta) and 3 (cyan, teal). The fused RON2L portion of the immunogen is shaded darker than the AMA1 portion. **b** Single-component immunogens superimposed on the AMA1-RON2L complex (PDB ID: 3zwz, white and orange). **c** A focused view of RON2L and the surrounding loops in single-component immunogens superimposed on the AMA1-RON2L complex (PDB ID: 3zwz). An arrow indicates local structural perturbations in the insertion fusion immunogens.

DII loop of AMA1 DI-DII and is consistent with the formation of an irreversibly bound RON2L complex.

## Functional antibody responses outside of the RON2L binding site contribute substantially to strain-transcending parasite neutralization

The neutralizing activity of immunogen-induced antibodies was evaluated in the GIA assay using day 63 sera. Purified total IgG from individual rats was first evaluated in the GIA assay against *Plasmodium falciparum 3D7*, which contains the same AMA1 and RON2 sequences used for design and vaccination. Antibodies from all groups, except the adjuvant only group, showed potent GIA in the range of 60–81% against *Plasmodium falciparum 3D7* (Fig. 5d). This demonstrates that the immunogens elicit a potent inhibitory antibody response similar to the AMA1 DI-DII and AMA1 DI-DII-RON2L complexes despite having drastically different RON2L in vitro blocking activity. We measured the $IC_{50}$ (the IgG concentration that gave 50% inhibition in GIA) of pooled IgG from each group to further quantify the GIA elicited by the immunogens. All groups had potent GIA against *Plasmodium*

*falciparum 3D7*, and there were insignificant differences in $IC_{50}$ among the AMA1 DI-DII, AMA1 DI-DII-RON2L complex and SBD1 immunogen groups (Fig. 6a, d, Supplementary Fig. 5a). However, the $IC_{50}$ values elicited by the insertion fusion immunogens 2 and 3 were 1.5-2-fold higher (i.e., less potent), and they were significantly different from that of the AMA1 DI-DII-RON2L complex ($P < 0.005$) (Fig. 6a, d, Supplementary Fig. 5a). Strikingly, when the same pooled IgGs were tested with *Plasmodium falciparum FVO* (Fig. 6b, e, Supplementary Fig. 5b) and *Plasmodium falciparum Dd2* (Fig. 6c, f, Supplementary Fig. 5c), only pooled IgGs from the AMA1 DI-DII-RON2L complex and SBD1 immunogen groups showed >50% inhibition at 5 mg/mL, and the $IC_{50}$ for the SBD1 immunogen group was significantly lower (i.e., more potent) than that for the AMA1 DI-DII-RON2L complex group in both strains ($P < 0.001$). The results suggest that either the local structural changes in domain I loops (Supplementary Fig. 2b, c and 6a, b) disrupt strain-transcending epitopes or that the epitopes within the disrupted DIf loop along with other conserved domain I loops are important for broad protection (Supplementary Fig. 6a, b). These results indicate the presence of functional epitopes on AMA1 DI-DII

**Table 1 | Data collection and refinement statistics**

|  | SBD1 immunogen (PDB ID: 8GID) | Insertion fusion immunogen 2 (PDB ID: 8GIE) | Insertion fusion immunogen 3 (PDB ID: 8GIF) |
|---|---|---|---|
| Data collection |  |  |  |
| Space group | C 1 2 1 | P 1 2₁ 1 | P 1 2₁ 1 |
| Cell dimensions |  |  |  |
| $a, b, c$ (Å) | 131.62 38.34 71.97 | 40.63 62.83 60.47 | 40.36 62.92 60.01 |
| α, β, γ (°) | 90.00 95.14 90.00 | 90.00 96.467 90.00 | 90.00 96.253 90.00 |
| Resolution (Å) | 19.45–1.8 (1.864 –1.8) | 19.22–1.85 (1.916–1.85) | 19.67– 2.101 (2.176–2.101) |
| $R_{merge}$ | 0.0701 (0.7521) | 0.0998 (0.6761) | 0.0686 (0.1663) |
| $R_{meas}$ | 0.0834 (0.8935) | 0.1189 (0.8021) | 0.0820 (0.2019) |
| Mean $I/\sigma(I)$ | 12.22 (2.02) | 10.61 (2.32) | 12.07 (5.41) |
| Completeness (%) | 97.30 (96.48) | 97.74 (99.22) | 98.27 (92.38) |
| Redundancy | 3.4 (3.4) | 3.4 (3.5) | 3.2 (2.8) |
| Refinement |  |  |  |
| Resolution (Å) | 19.45–1.8 | 19.22–1.85 | 19.67–2.101 |
| No. reflections | 32,734 | 25,339 | 17,237 |
| $R_{work}/R_{free}$ | 0.1785/0.2093 | 0.1915/0.2116 | 0.1696/0.2157 |
| No. atoms |  |  |  |
| Protein | 2692 | 2407 | 2466 |
| Ligand/ion | 0 | 0 | 0 |
| Water | 164 | 109 | 100 |
| B-factors (Å²) |  |  |  |
| Protein | 35.72 | 34.89 | 34.57 |
| Water | 38.30 | 38.93 | 35.28 |
| r.m.s. deviations |  |  |  |
| Bond lengths (Å) | 0.006 | 0.004 | 0.002 |
| Bond angles (°) | 0.79 | 0.63 | 0.48 |

Values in parentheses are for highest-resolution shell.

outside of its hydrophobic groove and DII loop that induced strain-transcending antibody responses that can neutralize diverse strains of malaria parasites.

## Discussion

The single-component SBD1 immunogen elicited potent GIA against all parasite strains tested. The IC$_{50}$ values against heterologous parasite strains *Plasmodium falciparum Dd2* and *FVO* were similar to the vaccine-matched strain *3D7*, suggesting that the antibodies elicited by SBD1 were broadly neutralizing. In our study, AMA1 alone elicited potent GIA against a vaccine-matched strain but not heterologous strains. Furthermore, the single-component SBD1 had significantly better strain-transcending GIA than the two-component AMA1-RON2L complex. These results are consistent with the shortcomings of past vaccine candidates and underscore the improvements offered by SBD1.

In addition to improving vaccine efficacy, structure-based immunogen design is a powerful approach to simplify antigen production for vaccine development. Here, we created three distinct immunogens with better production yields and biophysical characteristics that simplify the production of a single-component immunogen for use with a straightforward human relevant adjuvant for deployment in the field. We showed that blocking the domain I hydrophobic groove by fusing RON2L and removing the DII loop had no significant effect on the overall structure or immunogenicity. SBD1, in particular, is a single-component AMA1-RON2L immunogen that elicits GIA as potently or better than the two-component AMA1-RON2L complex. Thus, SBD1 appears to possess the most desirable characteristics for further development.

In addition to the development of a strain-transcending single-component immunogen, this study compared a vaccine induced antibody response that prevents RON2L binding versus a vaccine induced antibody response that targets segments of AMA1 independent of the RON2L binding site and DII loop. The data

showed that antibody responses induced by functional epitopes on AMA1 domains I and II outside the domain I hydrophobic groove are sufficient for effective neutralization of malaria parasites. These nonblocking epitopes of AMA1 DI-DII are poorly characterized and should be carefully explored to identify broadly protective epitopes for structure-guided design of even more potent immunogens and to provide new insights into parasite neutralization mechanisms.

All designed immunogens were structurally validated for accuracy. The three immunogens have a similar structure to the previously characterized AMA1 DI-DII-RON2L complex (PDB ID: 3zwz)[15]. A comparison of interface residues between both components of the designed immunogens and the previously described AMA1 DI-DII-RON2L complex (PDB ID: 3zwz) reveals very similar contacts. The DII loop shows high mobility in *Plasmodium falciparum*[32,64] and *Plasmodium vivax*[66] apo AMA1 structures and is stabilized by contacts with domain I. In apo AMA1, the DII loop covers a significant portion of the RON2L binding site[64,65]. However, the DII loop can be readily displaced to extend the hydrophobic groove to facilitate effective binding to RON2L[15], indicating that flexibility plays a critical role. Therefore, the DII loop was removed from our single-component immunogens, and the structures indicate that removal has no effect on binding between the RON2L and AMA1 components. A significant improvement in the stability of immunogens is likely due to binding between the RON2L and AMA1 DI-DII components, which not only enhances favorable interactions but also improves residue packing in domain I.

In previous studies, combining recombinant AMA1 with its ligand, RON2L, showed greater efficacy than AMA1 alone in the in vitro GIA assay and conferred enhanced protection in preclinical studies from a virulent challenge with *Plasmodium* parasites[62,63]. Additionally, this coimmunization resulted in higher levels of blocking antibodies than with AMA1 alone. We found no significant differences in blocking antibody levels among animals immunized with AMA1 DI-DII alone and the AMA1 DI-DII-RON2L complex. This may be due to variations in complex preparation, adjuvant, dose, animal model, and boundaries of immunogens between studies. However, we did see a significant increase in heterologous GIA against *Plasmodium falciparum FVO* and *Dd2* when RON2L was added to AMA1 as either a two-component or single-component SBD1 vaccine. This work provides valuable insights to facilitate the development of potent and durable interventions against malaria by utilizing structure-guided vaccine design[67–69]. Our results support the concept of an AMA1-RON2L vaccine and underscore the potential of SBD1.

There has been an intense focus on AMA1/RON2 blocking antibodies, but our data suggest that this focus should be broadened. Blocking antibodies may target epitopes within regions of AMA1 that serve to bind additional components of the RON complex[32–36]. The murine monoclonal antibody (mAb) 1F9 recognizes a strain-specific epitope within domain I of AMA1[32]. The 1F9 epitope shares a significant overlap with the epitopes of neutralizing single-domain antibodies 14I-1 and 14I1-M15[34] and strain-transcending human mAb humAbAMA1[36]. The invasion-inhibitory mAb 4G2 recognizes a strain-transcending epitope within the DII loop, prevents its displacement and possibly blocks interactions between AMA1 and RON2L necessary for recruitment into the MJ during invasion[35,70,71]. These RON2L blocking neutralizing epitopes are inaccessible in our designed immunogens due to the bound fused RON2L. Antibody responses elicited by our single-component immunogens did not block the binding between RON2L and AMA1, reflecting the inaccessibility of the domain I hydrophobic groove. However, elimination of these epitopes and antibodies did not significantly reduce the ability of the immunogens to induce neutralizing activity. Therefore, antibody responses elicited by our immunogens disrupt parasite growth via mechanisms other than

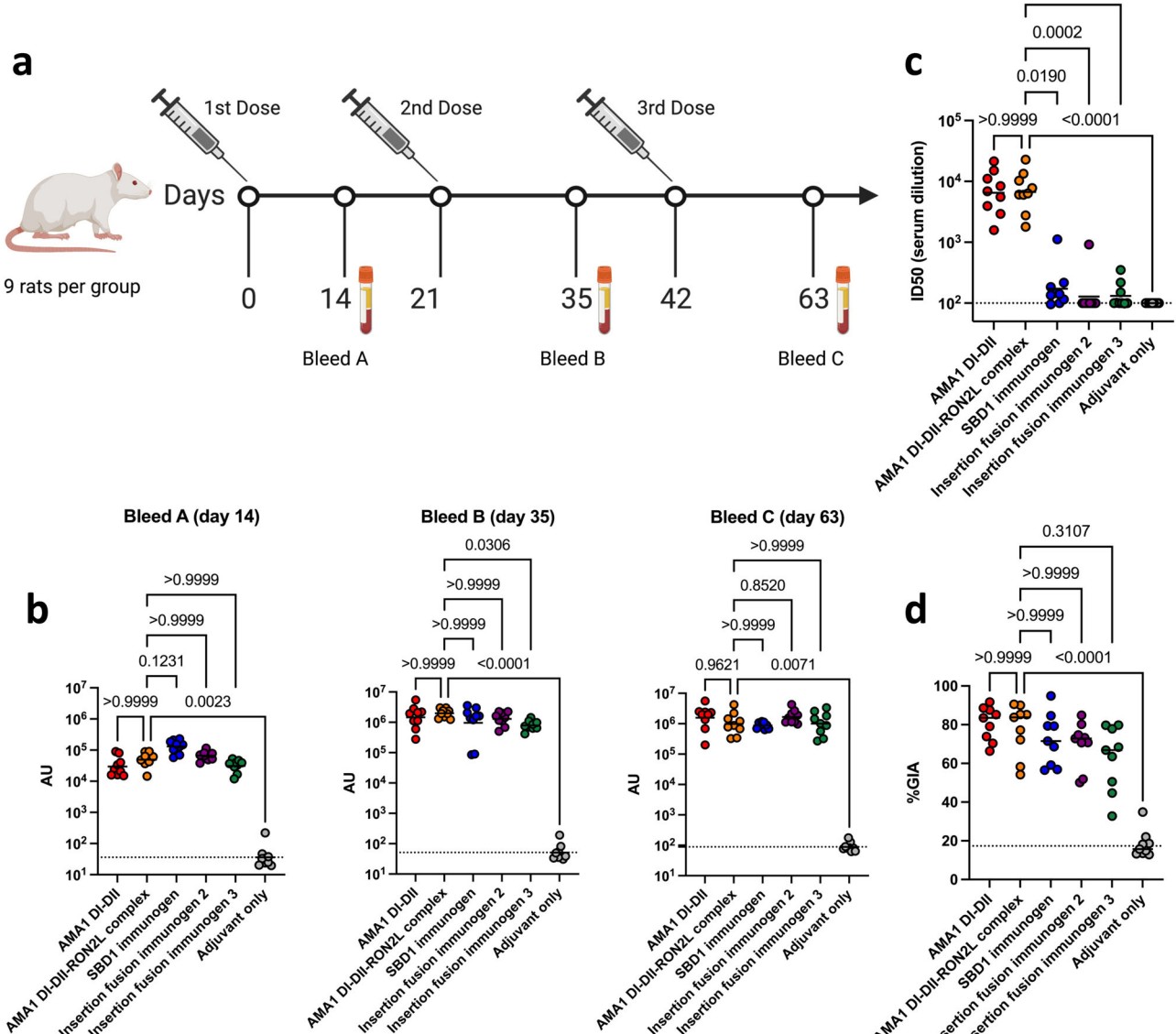

**Fig. 5 | Neutralizing antibody levels in rats immunized with single-component immunogens are similar to those of AMA1 DI-DII alone or the AMA1 DI-DII-RON2L complex. a** Immunization and blood draw scheme for rats. The figure was created with BioRender.com. **b** Serum IgG titers against AMA1 DI-DII from three independent biological replicates. The dashed line indicates the detection limit of the assay, and the bars represent the geometric mean titers (GMTs). **c** Serum antibody titers blocking the AMA1 DI-DII/RON2L interaction from two independent biological replicates depicted as described in (**b**). **d** In vitro GIA of purified IgG from individual rats from each group at day 63 was tested at 5.0 mg/ml against *Plasmodium falciparum 3D7* blood stage in three independent assays. Bars represent the median. The dashed line indicates the median of the adjuvant only group. Statistical comparisons and *P* values for (**b**), (**c**), and (**d**) were obtained using a Kruskal–Wallis analysis followed by Dunn's test to correct for multiple comparisons of the AMA1 DI-DII, immunogens and adjuvant only groups with the AMA1 DI-DII-RON2L complex group. Source data are provided as a Source data file.

RON2L receptor blockade and target epitopes on faces of AMA1 independent of the RON2L binding pocket.

Alternate neutralizing mechanisms have been identified for AMA1-specific antibodies, including impairing proteolytic processing and interfering with the redistribution of cleavage products[36,72]. Polyclonal rabbit IgG raised against the AMA1 ectodomain has been proposed to inhibit parasite invasion of erythrocytes by inhibiting secondary proteolytic processing of AMA1 and its redistribution. In addition, parasites that episomally express a shedding-resistant form of AMA1 were more sensitive to antibody-mediated parasite neutralization, suggesting that shedding of surface proteins during invasion helps the parasite evade host immunity[30]. The single-component immunogens are based on domains I and II of AMA1, and these domains are distant from the secondary cleavage site in AMA1. It is therefore unlikely that the

designs induce antibody responses that block secondary proteolytic processing, except when steric hindrance due to antibody size prevents processing. Additionally, an AMA1-specific neutralizing antibody isolated by phage display binds domain II without competing with RON2, suggesting an alternative mechanism for neutralizing parasites[73]. Thus, further studies are required to clearly define how antibody responses to AMA1 DI-DII outside of the hydrophobic groove/DII loop drive parasite neutralization.

In conclusion, the single-component SBD1 immunogen elicited potent strain-transcending growth inhibition that was significantly better than that elicited by AMA1 DI-DII alone or the AMA1 DI-DII-RON2L complex despite lacking receptor blocking antibodies that prevent RON2L binding to AMA1. Therefore, antibodies that target invasion-inhibitory epitopes outside the RON2L binding site have a

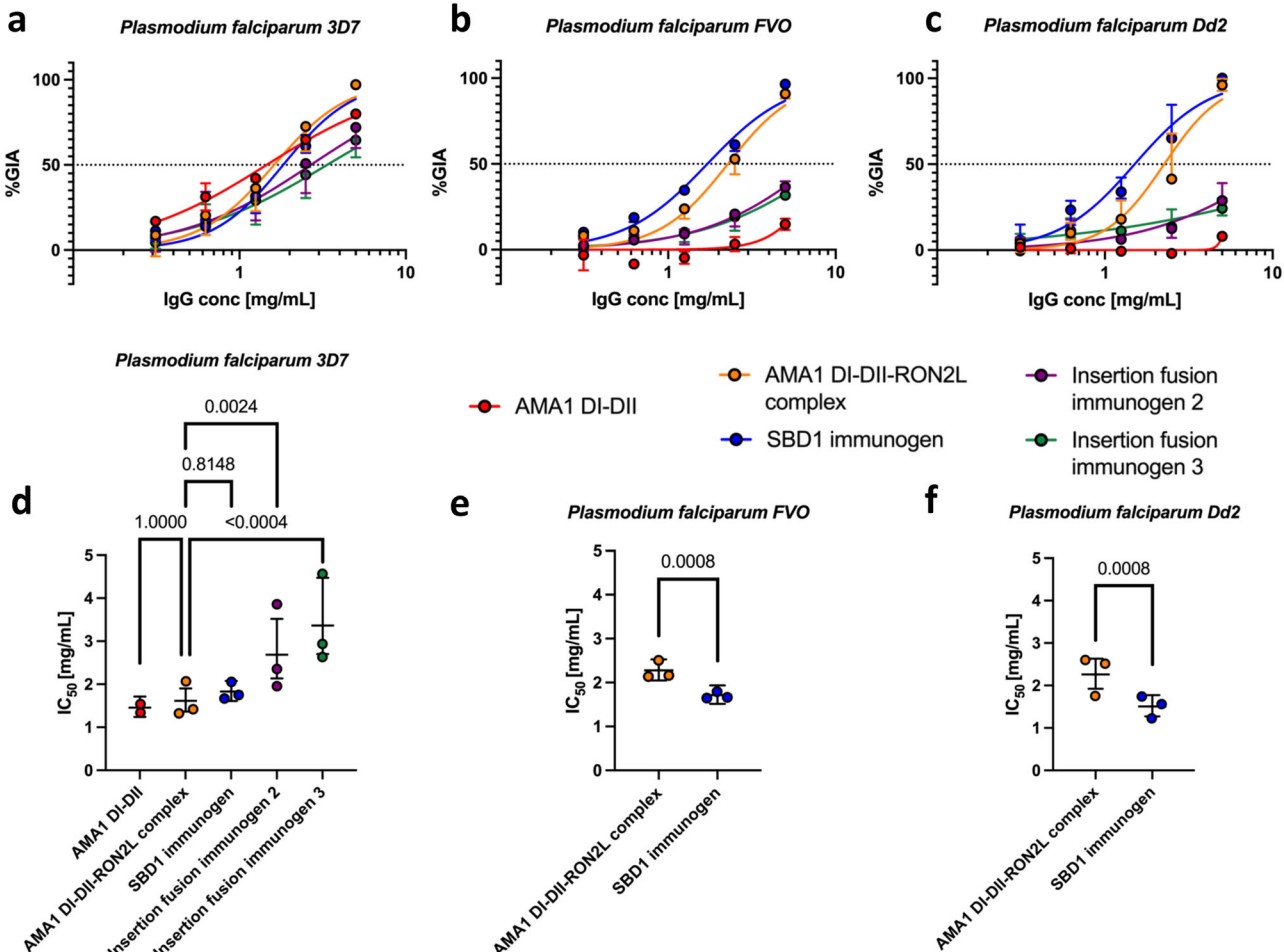

**Fig. 6 | SBD1 immunogen elicits significantly more potent strain-transcending antibodies than AMA1 DI-DII alone or the AMA1 DI-DII-RON2L complex.** In vitro GIA dilution series of pooled purified IgG from each group at day 63 against *Plasmodium falciparum* (**a**) *3D7* (**b**) *FVO* (**c**) *Dd2*. The data are plotted as the median with 95% CI and arise from three independent biological replicates for SBD1 immunogen, AMA1 DI-DII-RON2L Complex, insertion fusion immunogens 2 and 3, and two biological replicates for AMA1 DI-DII alone due to limited IgG for this group. Concentration (mg/ml) of pooled purified IgG required to demonstrate 50% inhibition ($IC_{50}$) against *Plasmodium falciparum* (**d**) *3D7* (**e**) *FVO and* (**f**) *Dd2* were determined by interpolation after fitting data globally to a four-parameter dose-response curve. The bars represent the $IC_{50}$ (center) and 95% CI of a global fit of three independent biological replicates (two biological replicates for the AMA1 DI-DII group). Points represent $IC_{50}$ values for individual fits of each biological replicate. Statistical comparisons were made using a two-tailed extra sum-of-squares *F*-test (with Bonferroni correction for **d**). Source data are provided as a Source data file.

profound ability to disrupt the growth of diverse parasite strains. Such strain-transcending invasion-inhibitory epitopes can be leveraged to improve the quality of protection. Similar to AMA1 alone, the insertion fusion immunogens 2 and 3 produced potent strain-specific responses but were ineffective in inducing a broad immune response, possibly due to the insertion of RON2L in the DIf loop. This insertion results in a loss of strain-transcending functional activity, suggesting that either the local structural perturbations introduced by the insertion fusion disrupt strain-transcending epitopes or epitopes within the DIf loop contribute to protection breadth. It should be noted that no other insertion fusion could be designed that also retains the DIf loop due to the need to place RON2L in close proximity to the binding site in AMA1. The single-component SBD1 immunogen simplifies manufacturing and provides a path to a cost-effective vaccine. The development of SBD1 and the discovery that it elicits strain-transcending protection provide insight into alternative mechanisms of parasite neutralization. These findings will contribute to the identification of novel strain-transcending epitopes and to the development of potent strain-transcending malaria and apicomplexan parasite interventions.

## Methods

### Expression and purification of AMA1 DI-DII and single-component immunogens

All three single-component immunogens, the *3D7* allele of AMA1 DI-DII, AMA1 DI-DII ΔDII-loop and AMA1 ectodomain, TrxA (thioredoxin), TrxA-RON2L fusions, and IgNAR 14I-1 were expressed in HEK293 cells, a system capable of post-translational modifications. The sequences for all constructs were codon optimized for expression in mammalian cells (GenScript), and all N-linked glycosylation sites (NXS/T) were modified by substituting the serine or threonine residue with an alanine residue to prevent glycosylation that is absent in endogenous *Plasmodium falciparum* proteins.

These optimized coding sequences for all three single-component immunogens, AMA1 DI-DII, and TrxA, were synthesized and cloned into a pHL-sec expression plasmid, which incorporates a 6xHis tag at the C-terminus, and transfected into Expi293F™ cells (Thermo Fisher Scientific, Cat# A14527) and grown according to the manufacturer's instructions. pHL-sec was a gift from Edith Yvonne Jones (Addgene plasmid # 99845; RRID:Addgene_99845)[74]. The soluble proteins were purified from cell-free supernatant four days post-transfection using Ni Sepharose™ Excel resin (Cytiva, Cat# 17371203) and size exclusion

chromatography (Superdex 200 Increase 10/300 GL; Cytiva) in phosphate buffered saline (PBS) (pH 7.4) or 20 mM Tris (pH 8.0) containing 100 mM NaCl. Size exclusion chromatography was performed on a ÄKTA pure protein purification system, and data were collected using UNICORN 7.3 software.

Purification yields of single-component immunogens were calculated as described previously[67]. Briefly, transfection, expression, and purification were performed in triplicate to determine the purification yields for single-component immunogens. A 100 ml culture was used for each replicate, and yields were calculated by integrating the area under the monomeric peak on the Abs280 chromatogram in size exclusion chromatography. These yields were similar to the yields obtained when fractions were pooled. The extinction coefficients were calculated from protein sequences using the ExPASy ProtParam tool[75] and used to calculate yields.

### Expression and purification of AMA1 ectodomain, IgNAR 14I-1, and TrxA-RON2L fusions

To produce the biotinylated AMA1 ectodomain and IgNAR 14I-1, the optimized coding sequence was synthesized and cloned into a derivative of the pHL-avitag3 expression plasmid, which incorporates an Avi-tag (GLNDIFEAQKIEWHE) and a 6xHis tag at the C-terminus (GenScript). pHL-avitag3 was a gift from Edith Yvonne Jones (Addgene plasmid # 99847; RRID:Addgene_99847)[74]. The plasmid was cotransfected with the BirA biotin ligase expressing plasmid and 100 μM biotin into Expi293F™ cells and grown according to the manufacturer's instructions. Secreted BirA-Flag was a gift from Gavin Wright (Addgene plasmid # 64395 ; RRID:Addgene_64395)[76]. The soluble biotinylated AMA1 ectodomain and IgNAR 14I-1 were purified from cell-free supernatant four days post-transfection using Ni Sephaose™ Excel resin (Cytiva) and size exclusion chromatography (Superdex 200 Increase 10/300 GL or Superdex 75 Increase 10/300 GL; Cytiva) in a buffer containing 10 mM HEPES (pH 7.4), 150 mM NaCl and 3 mM EDTA. Purified biotinylated AMA1 ectodomain and IgNAR 14I-1 were used for BLI experiments and bioassays. The expressed AMA1 ectodomain and IgNAR 14I-1 were biotinylated to at least 90%, as evidenced by SDS-PAGE gel-shift[77].

The TrxA-RON2L-1 fusion protein contains the loop region of RON2 (RON2L; residues Asp2021 to Ser2059) with N-terminal 6xHis and TrxA tags separated from the RON2L sequence by a PreScission Protease cleavage site (LEVLFQ/GP). A codon-optimized DNA sequence was synthesized and subcloned into a pHL-sec expression plasmid (GenScript). The plasmid was transfected into Expi293F™ cells and grown according to the manufacturer's instructions. Cell-free supernatant was harvested four days after transfection. The soluble TrxA-RON2L-1 fusion was purified using Ni Sepharose™ Excel resin (Cytiva) and size exclusion chromatography (Superdex 75 Increase 10/300 GL; Cytiva) in PBS (pH 7.4).

To produce the biotinylated TrxA-RON2L-2 fusion, a codon-optimized C-terminal Avi-tag (GLNDIFEAQKIEWHE) was appended to the TrxA-RON2L-1 sequence above, synthesized and subcloned into a pHL-sec expression plasmid (GenScript). The plasmid was cotransfected with the BirA biotin ligase expressing plasmid and 100 μM biotin into Expi293F™ cells and grown according to the manufacturer's instructions. The soluble biotinylated TrxA-RON2L-2 fusion was purified from cell-free supernatant four days post-transfection using Ni Sephaose™ Excel resin (Cytiva) and size exclusion chromatography (Superdex 75 Increase 10/300 GL; Cytiva) in a buffer containing 10 mM HEPES (pH 7.4), 150 mM NaCl and 3 mM EDTA. Purified biotinylated TrxA-RON2L-2 fusion was used for BLI experiments and bioassays. The expressed TrxA-RON2L-2 fusion was biotinylated to at least 90%, as evidenced by SDS-PAGE gel-shift[77]. Purified AMA1 ectodomain, IgNAR 14I-1, and TrxA-RON2L fusions (Supplementary Table 1) were of high purity and homogeneity (Supplementary Fig. 7).

### Preparation and purification of the AMA1 DI-DII-RON2L complex

To prepare the AMA1 DI-DII-RON2L complex, purified AMA1 DI-DII was mixed with purified TrxA-RON2L-1 fusion at a 1:2 molar ratio and incubated on ice for 30 min. The TrxA-RON2L-1 fusion contains a PreScission Protease cleavage site (LEVLFQ/GP) between the N-terminal TrxA/6xHis tags and RON2L (residues Asp2021 to Ser2059). A complex formed by mixing AMA1 DI-DII with TrxA-RON2L fusion proteins was proteolytically processed by PreScission Protease. Briefly, the sample was buffer exchanged and concentrated to 2 mg/ml (in 1 ml total volume) at 4 °C using an Amicon centrifugal filter (MilliporeSigma) equilibrated in cleavage buffer containing 50 mM Tris (pH 7.0), 150 mM NaCl, and 1 mM EDTA. The cleavage buffer did not contain reducing agents to avoid the reduction of intact disulfide bonds in AMA1 DI-DII and RON2L. Then, approximately 60 units of GST-tagged PreScission Protease was added and incubated at 4 °C for 5 h on a tube revolver (Thermo Fisher Scientific). One unit of PreScission Protease cleaves 100 μg of a test fusion protein in 16 h to 90% completion at 4 °C in cleavage buffer with 1 mM DTT. Following cleavage, the sample was applied to a column with a 1.5 ml bed volume of washed and equilibrated glutathione agarose resin (Gold Biotechnology, Cat# G-250) in cleavage buffer for removal of PreScission Protease. A flow-through fraction of the cleaved sample was collected and concentrated to 1.0 ml using an Amicon centrifugal filter (MilliporeSigma). The cleaved sample included the AMA1 DI-DII-RON2L complex, free uncomplexed RON2L and TrxA. The AMA1 DI-DII-RON2L complex from the cleaved sample was purified by size exclusion chromatography using a Superdex 75 Increase 10/300 GL column (Cytiva) equilibrated in PBS (pH 7.4) (Supplementary Fig. 8a). A peak containing AMA1 D-DII and RON2L confirms the formation of a stable complex and high purity (Supplementary Fig. 8b, c). Furthermore, the detection of RON2L by western blotting with the biotinylated AMA1 ectodomain as a probe confirmed that the disulfide bond in RON2L is intact, which is crucial for its binding to AMA1 (Supplementary Fig. 8c).

### Western Blotting

5 μg of AMA1 DI-DII-RON2L complex was diluted in Tricine SDS sample buffer (Thermo Fisher Scientific, Catalog# LC1676) without a reducing agent and incubated at room temperature for 5 min. Similarly, 2 μg of purified 6xHis-tagged AMA1 DI-DII and TrxA were diluted in 2x Tricine SDS sample buffer without reducing agent and used as controls. 10 μl of samples were loaded on a 16% Tricine gel (Thermo Fisher Scientific, Cat# EC66955BOX) and separated for 60 min at 150 volts. Proteins were transferred to a nitrocellulose (NC) membrane (Thermo Fisher Scientific, Cat# IB23002) using the iBlot™ Gel Transfer Device (Thermo Fisher Scientific) according to the manufacturer's instructions. The membrane was then washed three times with Tris buffered saline (20 mM Tris (pH 8.0), 150 mM NaCl) containing 0.1% Tween 20 (TBS/T) and blocked with 25 ml of 3% bovine serum albumin in TBS/T (blocking buffer) for 1 h at room temperature with gentle shaking and washed three times with TBS/T. The 6x-His Tag Monoclonal Antibody (Thermo Fisher Scientific, Cat# 37-2900) was diluted 1:10000 in 25 ml of blocking buffer and added to the membrane. The membrane was then incubated for 1 h at room temperature with gentle shaking and washed three times with TBS/T. The biotinylated AMA1 ectodomain was then diluted to 2 μg/ml in 25 ml of blocking buffer, added to the membrane, and incubated for 1 h at room temperature with gentle shaking, followed by three washes with TBS/T. Then, goat anti-mouse antibody conjugated to HRP (Jackson ImmunoResearch Laboratories Inc., Cat# 115-035-164) and streptavidin HRP conjugate (Thermo Fisher Scientific, Cat# 21127) were diluted 1:10000 and 1:5000, respectively, in 25 ml of blocking buffer, added to the membrane, incubated for 1 h at room temperature with gentle shaking, and washed three times with TBS/T. Next, chemiluminescent substrate (Thermo Fisher Scientific, Cat# 34579) was applied to the

membrane according to the manufacturer's instructions, and captured the chemiluminescence image using Amersham™ Imager 600 (GE Healthcare).

## Binding kinetics of AMA1 DI-DII and single-component immunogens with IgNAR 14I-1 or RON2L using biolayer interferometry

Binding of the AMA1 DI-DII and single-component immunogens to the IgNAR 14I-1 and RON2L were measured by kinetic experiments carried out on an Octet RED96e (Sartorius). All constructs were buffer exchanged into 1x HBS-EP+ buffer [10 mM HEPES (pH 7.4), 150 mM NaCl, 3 mM EDTA, and 0.05% (v/v) P20 surfactant (Cytiva, Cat# BR100826)] using Zeba™ spin desalting columns (Thermo Fisher Scientific) according to the manufacturer's instructions. All measurements were performed at 200 μl/well in 1x HBS-EP+ buffer at 25 °C in 96-well black plates (Greiner Bio-One, Cat# 655209). Streptavidin (SA) biosensors (Sartorius, Cat# 18-5019) were used to immobilize biotinylated IgNAR 14I-1 [~0.6 binding (nm) units] or TrxA-RON2L [~0.3 binding (nm) units] for 300 s. Immunogens were two-fold serially diluted in HBS-EP+ buffer in the range of 200 nM to 3.125 nM. Assay was performed in five sequential steps with Octet® BLI Discovery 12.2.2.20 software (Sartorius): Step 1, biosensor hydration and equilibration (780 s); Step 2, immobilization of biotinylated IgNAR 14I-1 or TrxA-RON2L on a Streptavidin (SA) biosensor (300 s); Step 3, wash and establish baseline (60 s); Step 4, measure IgNAR 14I-1 or TrxA-RON2L-immunogens association kinetics (300 s); and Step 5, measure IgNAR 14I-1 or RON2L-immunogens dissociation kinetics (300 s). The acquired raw data for the binding of AMA1 DI-DII with IgNAR 14I-1 or RON2L were processed and globally fit to a 1:1 binding model with Octet® Analysis Studio 12.2.2.26 Software (Sartorius). The binding kinetics measurements were carried out in three replicates. Values reported are the average and SEM among replicates.

## Binding analysis of AMA1 DI-DII and single-component immunogens with IgNAR 14I-1 or RON2L using ELISA

Binding of the AMA1 DI-DII and single-component immunogens to the IgNAR 14I-1 and RON2L were analyzed by ELISA. Immunogens were diluted in 50 mM Na-carbonate (pH 9.5) and coated on Nunc MaxiSorp flat-bottom 96-well ELISA plates (Thermo Fisher Scientific, Cat# 44-2404-21) at 10 nM in 100 μl at 4 °C overnight. The plates were then washed three times with PBS containing 0.05% Tween 20 (PBS/T), blocked with 2% bovine serum albumin in PBS/T for 1 h at room temperature, and then washed three times with PBS/T. Next, 200 μl of biotinylated 14I-1 or TrxA-RON2L-2 diluted to 200 nM and 1000 nM in blocking buffer (PBS/T with 2% bovine serum albumin) was added to each well of the blocked plates, incubated for 1 h at room temperature, and then washed three times with PBS/T. Then, 200 μl of streptavidin HRP conjugate (Thermo Fisher Scientific, Cat# 21127) was added to each well at a 1:10000 dilution and incubated for 1 h at room temperature. The plates were then washed three times with PBS/T and developed with 70 μl of TMB substrate (MilliporeSigma) for 20 min at room temperature in the dark. The reaction was then stopped by adding 160 mM sulfuric acid ($H_2SO_4$), and the absorbance was measured at 450 nm on a BioTek™ Synergy H1 microplate reader using Gen5 3.08.01 software.

## Differential scanning fluorimetry

Differential scanning fluorimetry was performed to measure the thermal stability of single-component immunogens using the Protein Thermal Shift™ Dye Kit (Thermo Fisher Scientific, Cat# 4461146) according to the manufacturer's instructions. Each 20 μl assay mixture contained 10 μg of purified immunogen in PBS (pH 7.4), 1× Protein Thermal Shift buffer, and 1× Thermal Shift Dye. The melt-curve experiments were performed on a 7500 Fast Real-Time polymerase chain reaction system (Thermo Fisher Scientific). Fluorescence readings were monitored as the temperature was increased from 25 to 95 °C at a ramp rate of 1%. Protein melt fluorescent readings were analyzed using Protein Thermal Shift™ software v 1.4 (Thermo Fisher Scientific), and the melting temperature ($T_m$) was calculated as a peak of the derivative melt curve. Protein melt-curve experiments were performed in five technical replicates on each plate and in biological triplicate. $T_m$ for a biological replicate was calculated by averaging technical replicates, and the reported $T_m$ was calculated by averaging three biological replicates.

## Protein crystallization, data collection, and structure solution

6xHis-tagged immunogens were purified from cell-free supernatant by affinity chromatography using Ni Sepharose™ Excel resin (Cytiva, Cat# GE17371201) according to the manufacturer's instructions followed by size exclusion chromatography using a Superdex 200 Increase 10/300 GL column (Cytiva) equilibrated in 20 mM Tris (pH 8.0) and 100 mM NaCl. Purified immunogens were concentrated to 20 mg/ml using an Amicon centrifugal filter (MilliporeSigma). Crystallization experiments were carried out using hanging drop vapor diffusion. Crystals were obtained using a mosquito® crystal (SPT Labtech) to mix 0.2 μl of purified immunogen (20.0 mg/ml) with 0.2 μl reservoir solution in 96-well plates that were incubated at 18 °C. SBD1 immunogen was crystallized with 0.2 M ammonium sulfate and 20% (w/v) polyethylene glycol (PEG) 3350 at 18 °C. Insertion fusion immunogen 2 was crystallized with 0.5 M lithium chloride, 0.1 M Tris (pH 8.5), and 34% (w/v) PEG 6000 at 18 °C. Insertion fusion immunogen 3 was crystallized with 0.2 M magnesium chloride, 0.1 M Tris (pH 8.5), and 20% (w/v) PEG 8000 at 18 °C. All crystals were cryoprotected with the addition of either 30% glycerol or 30% PEG 400 and flash-frozen in liquid nitrogen. Diffraction data for all crystals were collected at 1.0 Å at 100 K on the beamline SER-CAT 22-ID at the Advanced Photon Source (APS). All diffraction data were processed using XDS[78]. Reflections were indexed and integrated using XDS[78]. Data were scaled and merged using XSCALE[78] or POINTLESS and AIMLESS[79] and all structures were solved by molecular replacement (MR) using Phaser[80–82], rebuilt with AutoBuild[82,83], manually built in Coot[84], and refined with Phenix.refine[82,85]. Resolution cutoffs for scaling were evaluated using standard metrics of signal to noise and CC½. Standard settings in Phenix.refine, TLS parameters[86], B-factors, and weight optimization options (X-ray/stereochemistry weight and X-ray/ADP weight) were enabled for the refinement of the immunogens. The crystal structures of all three immunogens were solved by MR using the AMA1-RON2L peptide complex (PDB ID: 3zwz) as a search model. Following final refinement, the $R_{work}/R_{free}$ values for the SBD1 immunogen, insertion fusion immunogen 2, and insertion fusion immunogen 3 were 0.1737/0.2076, 0.1915/0.2116, and 0.1696/0.2157, respectively. MolProbity was used to evaluate the geometry of the final models[87,88]. All three immunogens showed more than 96.0% of the residues as Ramachandran favored and 0% outlier residues. Figures of molecular structures were generated using the PyMOL Molecular Graphics System, Version 2.5 (Schrödinger, Inc.). The software used in this project was curated by SBGrid[89].

## Rat immunizations

Rat immunogenicity studies were performed in an American Association for Accreditation of Laboratory Animal Care-accredited facility under the guidelines and approval of the Institutional Animal Care and Use Committee (approved protocol number: LMIV 1E) at the National Institutes of Health. On Day 0, groups of nine 12–14-week-old CD® (Sprague Dawley) IGS female rats, Crl:CD(SD) (Charles River Laboratories), were immunized by subcutaneous injection with 20 μg of each antigen in 100 μL formulated as a 1:1 volume ratio in AddaSO3™ adjuvant (InvivoGen, Cat# vac-as03-10) and DPBS (pH 7.4). Rats were

boosted twice after the initial prime on days 21 and 42. On days 14, 35, and 63, blood was collected, and serum was separated and stored at −80 °C.

## Serum antibody titer ELISA

The *Plasmodium falciparum 3D7* allele of AMA1 DI-DII was diluted in 50 mM Na-carbonate (pH 9.5) and coated on Nunc MaxiSorp flat-bottom 96-well ELISA plates (Thermo Fisher Scientific, Cat# 44-2404-21) at 20 µg/ml in 100 µl at 4 °C overnight. The plates were then washed three times with PBS containing 0.05% Tween 20 (PBS/T), blocked with 2% bovine serum albumin in PBS/T for 1 h at room temperature, and then washed three times with PBS/T. Next, serum was diluted in blocking buffer (PBS/T with 2% bovine serum albumin), and 100 µl was added to each well, incubated for 1 h at room temperature, and then washed three times with PBS/T. Then, 200 µl of goat anti-rat antibody conjugated to horseradish peroxidase (HRP) (secondary, Jackson ImmunoResearch Laboratories Inc., Cat# 112-035-071) was added to each well at a 1:5000 dilution and incubated for 1 h at room temperature. The plates were then washed three times with PBS/T and developed with 70 µl of 3,3′,5,5′-tetramethylbenzidine (TMB) substrate (MilliporeSigma, Cat# T0440-1L) for 20 min at room temperature in the dark. The reaction was then stopped by adding 2 M sulfuric acid ($H_2SO_4$), and the absorbance was measured at 450 nm on a BioTek™ Synergy H1 microplate reader using Gen5 3.08.01 software.

The reference standard curve was prepared using pooled serum from rats as described previously[67]. Pooled serum from rats immunized with AMA1 DI-DII and having relatively high antibody titers was used as a reference standard curve on each plate to determine the antibody titers of individual animals in all groups. The dilution of reference standard serum required to achieve an Abs450 value of 1 was defined as one antibody unit (AU). Three replicates of twofold serial dilutions of reference standard serum ranging from 20 to 0.01 AU were included in each plate. Serum from each animal was diluted such that the Abs450 value fell within the dynamic range of the reference standard curve. The Abs450 values for the reference standard curve were fitted to a four-parameter logistic curve to convert the Abs450 values into AUs for individual animals in all groups. AUs for each individual animal were measured in three replicates on separate plates, and an average was calculated and reported.

## AMA1 DI-DII/RON2L blocking assay

The AMA1 DI-DII/RON2L-blocking assay was carried out similarly to that described previously[67]. TrxA-RON2L-1 fusion was diluted in 50 mM Na-carbonate (pH 9.5) and coated on Nunc MaxiSorp flat-bottom 96-well ELISA plates (Thermo Fisher Scientific, Cat# 44-2404-21) at 20 µg/ml in 100 µl at 4 °C overnight. The plates were then washed three times with PBS containing 0.05% Tween 20 (PBS/T), blocked with 2% bovine serum albumin in PBS/T for 1 h at room temperature, and then washed three times with PBS/T. Next, serum was diluted in blocking buffer (PBS/T with 2% bovine serum albumin) in a twofold dilution series ranging from 1:50 to 1:6400. A total of 110 µl of diluted serum was mixed with 110 µl of 0.2 nM biotinylated AMA1 ectodomain or 100 µl of buffer as a background control and incubated for 1 h at room temperature. 200 µl of serum mixture was added to each well of the blocked plates, incubated for 1 h at room temperature, and then washed three times with PBS/T. Then, 200 µl of streptavidin HRP conjugate (Thermo Fisher Scientific, Cat# 21127) was added to each well at a 1:10000 dilution and incubated for 1 h at room temperature. The plates were then washed three times with PBS/T and developed with 70 µl of TMB substrate (MilliporeSigma) for 20 min at room temperature in the dark. The reaction was then stopped by adding 160 mM sulfuric acid ($H_2SO_4$), and the absorbance was measured at 450 nm on a BioTek™ Synergy H1 microplate reader using Gen5 3.08.01 software.

AMA1DI-DII/RON2L binding inhibition was determined by subtracting the Abs450 values from background controls lacking the biotinylated AMA1 ectodomain. The average maximum signal was calculated using three wells without serum. The following formula was used to calculate inhibition.

$$\%\text{Inhibition} = 100 \times (1 - X/\text{max})$$

where **X** is the Abs450 value of a well after background subtraction and **max** is the average value of the three wells without serum after background subtraction.

The blocking assay was performed in duplicate, percent inhibition values were calculated for each serum dilution, and average values were plotted in GraphPad Prism 8. Data were fitted using a normalized dose response curve with a variable slope.

$$Y = 100/(1 + (ID50/X) \wedge \text{HillSlope})$$

where **X** is the serum dilution, **Y** is the % inhibition, and **HillSlope** and **ID50** are calculated parameters corresponding to the slope of the curve and the dilution at which 50% inhibition occurs, respectively. For each animal, the **ID50** values were plotted alongside the geometric mean value for each group.

## Growth inhibition assay (GIA)

All assays for GIA were performed as described in the protocol of the International Growth Inhibition Assay Reference Center at the National Institutes of Health[90]. IgG was purified from individual rat serum using Protein G HTC Agarose resin/Protein G Sepharose 4 Fast Flow resin (GoldBio, Cat# P-430-25 or Cytiva, Cat# 17061805) according to the manufacturer's instructions. Purified IgG was buffer exchanged in RPMI 1640 and concentrated with Amicon centrifugal filters (MilliporeSigma) to 10 mg/ml, and aliquots were stored at −80 °C. For the individual GIA, test IgG was added to triplicate wells at 5.0 mg/ml and incubated with infected red blood cells (*3D7* strain, 0.3% parasitemia, 1% hematocrit) in a final volume of 40 µl and returned to a culture incubator (5% $O_2$–5% $CO_2$–90% $N_2$) for 40 h at 37 °C. Growth inhibition (parasitemia) was assessed by the lactate dehydrogenase activity assay. The percent GIA was calculated using the following formula: % GIA = 100−100 (sample $A_{650}$ − uninfected RBC $A_{650}$)/(infected control $A_{650}$ − uninfected RBC $A_{650}$).

## Statistical Analyses

The serum IgG titers against AMA1 DI-DII were compared using a Kruskal–Wallis analysis followed by Dunn's test to correct for multiple comparisons of the AMA1 DI-DII, immunogens and adjuvant only groups with the AMA1 DI-DII-RON2L complex group. The ID50 values obtained in the AMA1 DI-DII/RON2L blocking assay were compared by a Kruskal–Wallis test followed by Dunn's test to correct for multiple comparisons of the AMA1 DI-DII, immunogens and adjuvant only groups with the AMA1 DI-DII-RON2L complex group. The %GIA levels among different groups were compared by a Kruskal–Wallis test followed by Dunn's test. For the strain-transcending GIA, a pooled IgG from each group was tested at 2-fold serial dilutions (5 to 0.313 mg/mL) with *3D7*, *FVO* or *Dd2* strains of parasites in the same way. The $IC_{50}$ value of each pooled IgG for each strain was calculated with data from 3 independent biological replicates (2 biological replicates for the AMA1 DI-DII group) using a 4-parameter sigmoidal fit, and then $IC_{50}$ values between different groups were compared by a two-tailed extra sum-of-squares *F*-test. When more than two groups were compared, the Bonferroni correction was performed.

## Reporting summary

Further information on research design is available in the Nature Portfolio Reporting Summary linked to this article.

## Data availability

All data generated or analyzed during this study are included in this published article, source data file and supplementary information files. Atomic coordinates and structure factors have been deposited in the Protein Data Bank with PDB IDs 8GID, 8GIE, and 8GIF. Source data are provided with this paper.

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

## Acknowledgements

This work was supported by the Intramural Research Program of the Division of Intramural Research, National Institute of Allergy and Infectious Diseases (NIAID), National Institutes of Health (NIH). The GIA activity was also supported by an Interagency Agreement (AID-GH-T-15-00001) with USAID's Malaria Vaccine Development Program. The findings and conclusions in this report are those of the author(s) and do not necessarily represent the official position of the U.S. Agency for International Development. Data were collected at Southeast Regional Collaborative Access Team (SER-CAT) 22-ID beamline at the Advanced Photon Source, Argonne National Laboratory. SER-CAT is supported by its member institutions, and equipment grants (S10_RR25528, S10_RR028976 and S10_OD027000) from the NIH. Use of the Advanced Photon Source was supported by the U. S. Department of Energy, Office of Science, Office of Basic Energy Sciences, under Contract No. W-31-109-Eng-38. This study used the Office of Cyber Infrastructure and Computational Biology (OCICB) High Performance Computing (HPC) cluster at the National Institute of Allergy and Infectious Diseases (NIAID), Bethesda, MD.

## Author contributions

N.H.T. conceived the study. N.H.T., T.H.D. and P.N.P. conceived the immunogen design. N.H.T. and P.N.P. conceived structural studies, biophysical characterization, binding analyses, serum antibody titer ELISA, and blocking assay. N.H.T., P.N.P., K.M. and C.A.L. conceived the functional analysis of serum IgG. N.H.T., P.N.P., N.D.S. and L.E.L. conceived the animal studies. P.N.P., T.H.D., A.D., N.D.S., K.M., H.M. and T.O. performed experiments and analyzed the data. N.H.T., L.E.L., K.M. and C.A.L. supervised the studies and analyzed the data. P.N.P. and N.H.T wrote the manuscript, with input from all authors.

## Competing interests

N.H.T., T.H.D., and P.N.P. are listed as inventors on a provisional patent application related to this work. The remaining authors declare no competing interests.
