## [Peer Review File · Nature Communications]

Structure-based design of a strain transcending AMA1-RON2L malaria vaccineReviewers' Comments:

Reviewer #1:

Remarks to the Author:

In this study, the authors describe the successful of three single-component AMA1-RON2L protein antigens, based on previous work which has indicated that immunisation with AMA1 + RON2L complexes raise parasite-neutralising antibody titers. This manuscript is extremely well written: I commend the authors on the clarity of the writing and the easy-to-follow presentation of the results.

In addition to the description of the novel constructs, a number of claims regarding the biophysical nature, immunogenicity and protective capacity of the immunogenicity are made. A selection of the most notable claims made in this study are:

1. IgNAR14I-1 binding measurements revealed that the RON2L binding site of AMA-1 within all three constructs is not available for binding. This is well-supported by the results displayed in figure 3A&B.
2. The structures of the designed immunogenicity are similar to that of the AMA1 D1/D2-RON2L complex, with SBD1 being most similar and the insertions fusions showing structural differences in loops near the insertion sites. Figure 4 shows these similarities qualitatively, and RMSDs are provided in the text to quantify the similarity. Would it be possible to plot the variation on a positional or matched residue-residue basis (e.g. scatterplot or line plot) to show the differences across the length of the construct quantitatively?
3. The designed immunogenicity had similar immunogenicity to apo AMA1-DI-DII and AMA1/RON2L complex. This is supported by figure 5b. In the accompanying text (L254) it is stated that there were no significant differences, though I cannot see detail of any statistical tests being performed.
4. Antibodies raised to all three immunogens elicited very low or negligible titers of antibodies that disrupt AMA1-RON2L binding. This is well supported by the assay shown in figure 5c. This is a substantial and interesting finding, given the next point.
5. Despite low or undetectable levels of antibodies that inhibit AMA1-RON2L binding, antibodies elicited to the three fusion immunogens have high levels of growth inhibition measures on GIA against Pf 3D7.
6. The SBD1 immunogen elicits antibodies that inhibit Pf FVO and DD2 strains, with a more potent inhibitory activity than antibodies raised against AMA1-RON2L complex. This is clearly demonstrated by the data shown in fig 6b, c, e and f.

Overall the paper's findings are well supported by the data presented and those findings are of substantial interest to those within the field of malaria vaccinology. The general principle of the approach and the broad findings are also likely to be of interest more widely within the field of vaccine development.

The methods described are sufficiently detailed to be reproduced. I have made a note below of a couple of minor comments regarding the figures:

Figure legends - the description of some panels could be fuller to aid the reader. E.g. 1e could include information on the technique used. In the figure 5 legend, the abbreviation GMT is used, but not defined (presumably geometric mean titer).

ELISA (figure 5) - I cannot see a section defining how 'AB units' are defined (should this be expressed as AU - this is defined in methods)

L449 (methods) - remove hyphen in 100-ml

Reviewer #2:

Remarks to the Author:

The manuscript by Patel et al., designed and characterized structure based chimeric immunogens of the AMA1-RON2L complex, incorporating the RON2L peptide into the AMA1 sequence by either circular permutation (SBD1) or through deletion of the D1f loop and insertion. These immunogens showed good biochemical properties, blocked RON2L binding, and crystal structures showed that they mimicked the AMA1-RON2L complex well. The authors then performed immunization studies, with all immunogens eliciting high antibody titers and neutralizing activity. Importantly however, only the SBD1 immunogen and the AMA1-RON2L complex were capable of eliciting strain-transcending neutralizing activity.

Collectively, along with the Yanik manuscript, these are well-designed and important studies for the malaria vaccine field, utilizing structural biology to develop novel immunogens, and should be of interest to the community. The crystal structures are well described and of excellent quality. The immunization studies were thorough and clearly presented and the findings impressive.

Major Comments

In the discussion you suggest that antibodies to conserved loops surrounding the RON2L binding site are likely the target for strain transcending antibodies. Did the authors test for antibodies to RON2L in the sera of rats immunized with AMA1-RON2L complex or the chimeric immunogens? Since RON2L is conserved in the strains tested, couldn't it also be a source for strain transcending antibodies? And if antibodies to RON2L are elicited, and if you deplete them, are the anti-AMA1 antibodies elicited by SBD1 still inhibitory and strain transcending? This would rule in/out RON2L as a target for strain transcending antibodies and provide further evidence that rational design of an AMA1-RON2L immunogen should focus on eliciting strain-transcending antibodies to highly conserved loops on AMA1.

Minor Comments

Line 159 - remove 'at'

Line 615 - Was it polyethylene glycol or ethylene glycol that you used to cryoprotect?

Line 783 - In the BLI in Fig. 3 what was the top concentration of analyte used?

Perhaps a supplementary figure with a sequence alignment of 3D7, FVO and Dd2 AMA1 sequences would be beneficial, as well as mapping polymorphisms onto the AMA1 structure with a particular focus on the loops?

Reviewer #1 (Remarks to the Author):

Comment: In this study, the authors describe the successful of three single-component AMA1-RON2L protein antigens, based on previous work which has indicated that immunisation with AMA1 + RON2L complexes raise parasite-neutralising antibody titers. This manuscript is extremely well written: I commend the authors on the clarity of the writing and the easy-to-follow presentation of the results.

Response: We thank the reviewer for their positive comments.

Comment: In addition to the description of the novel constructs, a number of claims regarding the biophysical nature, immunogenicity and protective capacity of the immunogenicity are made. A selection of the most notable claims made in this study are:

Response: We thank the reviewer for concisely summarizing the key findings and offering insightful comments. In response, we have carefully addressed each point raised, providing a comprehensive point-by-point response below.

Comment: 1. IgNAR14I-1 binding measurements revealed that the RON2L binding site of AMA-1 within all three constructs is not available for binding. This is well-supported by the results displayed in figure 3A&B.

Response: We thank the reviewer for their positive comments.

Comment: 2. The structures of the designed immunogenicity are similar to that of the AMA1 D1/D2-RON2L complex, with SBD1 being most similar and the insertions fusions showing structural differences in loops near the insertion sites. Figure 4 shows these similarities qualitatively, and RMSDs are provided in the text to quantify the similarity. Would it be possible to plot the variation on a positional or matched residue-residue basis (e.g. scatterplot or line plot) to show the differences across the length of the construct quantitatively?

Response: We thank the reviewer for identifying ways to improve the manuscript. In the revised manuscript, we have included a supplementary figure that illustrates root-mean-square deviation (RMSD) data for the C α atom of each residue in all three immunogens, when aligned to the AMA1-RON2L complex. We have provided this data in Supplementary Figure 2 and incorporated corresponding changes into the text. This revision further supports the inferences and conclusions presented to demonstrate that SBD1 is much more similar to the AMA1 RON2L complex than the insertion fusions.

Comment: 3. The designed immunogenicity had similar immunogenicity to apo AMA1-DI-DII and AMA1/RON2L complex. This is supported by figure 5b. In the accompanying text (L254) it is stated that there were no significant differences, though I cannot see detail of any statistical tests being performed.

Response: We apologize for not including statistical comparisons and p -values for ELISA data in Figure 5b. We have updated Figure 5b in the revised manuscript to illustrate statistical comparisons and p -values. In Figure 5 legend and “Statistical Analyses” methods section, we have provided information about the statistical tests conducted. This revision further supports the inferences and conclusions presented that the designed immunogens had similar immunogenicity to AMA1-DI-DII and the AMA1/RON2L complex.

Comment: 4. Antibodies raised to all three immunogens elicited very low or negligible titers of antibodies that disrupt AMA1-RON2L binding. This is well supported by the assay shown in figure 5c. This is a substantial and interesting finding, given the next point.

Response: We thank the reviewer for their positive comments.

Comment: 5. Despite low or undetectable levels of antibodies that inhibit AMA1-RON2L binding, antibodies elicited to the three fusion immunogens have high levels of growth inhibition measures on GIA against Pf 3D7.

Response: We thank the reviewer for their positive comments.

Comment: 6. The SBD1 immunogen elicits antibodies that inhibit Pf FVO and DD2 strains, with a more potent inhibitory activity than antibodies raised against AMA1-RON2L complex. This is clearly demonstrated by the data shown in fig 6b, c, e and f.

Response: We thank the reviewer for their positive comments.

Comment: Overall the paper's findings are well supported by the data presented and those findings are of substantial interest to those within the field of malaria vaccinology. The general principle of the approach and the broad findings are also likely to be of interest more widely within the field of vaccine development.

Response: We thank the reviewer for their positive comments.

Comment: The methods described are sufficiently detailed to be reproduced. I have made a note below of a couple of minor comments regarding the figures:

Response: We thank the reviewer for their positive comments and for identifying ways to improve the manuscript.

Comment: Figure legends - the description of some panels could be fuller to aid the reader. E.g. 1e could include information on the technique used.

Response: We thank the reviewer for identifying ways to improve the manuscript. The design process that was used to develop the immunogens shown in Fig. 1e are described individually in 1b, 1c and 1d and we do not believe it would aid the reader to repeat this information in 1e.

Comment: In the figure 5 legend, the abbreviation GMT is used, but not defined (presumably geometric mean titer).

Response: We apologize that GMT was not clearly defined and GMT has been defined on first usage in the revised manuscript. We have updated the figure legends to include the necessary details.

Comment: ELISA (figure 5) - I cannot see a section defining how 'AB units' are defined (should this be expressed as AU - this is defined in methods)

Response: We thank the reviewer for identifying an error in the manuscript. In the revised submission, we have corrected this error to AU in Figure 5b.

Comment: L449 (methods) - remove hyphen in 100-ml

Response: In line 449, the hyphen is removed in the revised manuscript.

Reviewer #2 (Remarks to the Author):

Comment: The manuscript by Patel et al., designed and characterized structure based chimeric immunogens of the AMA1-RON2L complex, incorporating the RON2L peptide into the AMA1 sequence by either circular permutation (SBD1) or through deletion of the D1f loop and insertion. These immunogens showed good biochemical properties, blocked RON2L binding, and crystal structures showed that they mimicked the AMA1-RON2L complex well. The authors then performed immunization studies, with all immunogens eliciting high antibody titers and neutralizing activity. Importantly however, only the SBD1 immunogen and the AMA1-RON2L complex were capable of eliciting strain-transcending neutralizing activity.

Response: We thank the reviewer for concisely summarizing the key findings and offering insightful comments.

Comment: Collectively, along with the Yanik manuscript, these are well-designed and important studies for the malaria vaccine field, utilizing structural biology to develop novel immunogens, and should be of interest to the community. The crystal structures are well described and of excellent quality. The immunization studies were thorough and clearly presented and the findings impressive.

Response: We thank the reviewer for their positive comments.

Comment: Major Comments

In the discussion you suggest that antibodies to conserved loops surrounding the RON2L binding site are likely the target for strain transcending antibodies. Did the authors test for antibodies to RON2L in the sera of rats immunized with AMA1- RON2L complex or the chimeric immunogens? Since RON2L is conserved in the strains tested, couldn't it also be a source for strain transcending antibodies? And if antibodies to RON2L are elicited, and if you deplete them, are the anti-AMA1 antibodies elicited by SBD1 still inhibitory and strain transcending? This would rule in/out RON2L as a target for strain transcending antibodies and provide further evidence that rational design of an AMA1-RON2L immunogen should focus on eliciting strain-transcending antibodies to highly conserved loops on AMA1.

Response: We thank the reviewer for identifying the next phase of study for this work. The study demonstrates that SBD1 provides strain-transcending protection. However, the precise structural mechanism for how the strain-transcending protection is manifested requires further extensive study that is beyond the scope of the current manuscript. The mechanism for strain transcending protection is likely complex and will have contributions from AMA1, RON2L and the conformation of the complex. Identifying which of these is critical will require careful and extensive study.

For example, the reviewer inquires if RON2L conservation could be the driving force for strain-transcending protection. It is important to note that all of our immunogens and

AMA1-RON2L complex include RON2L. Despite the presence of the highly conserved RON2L in all immunogens, only SBD1 immunogen and AMA1-RON2L complex exhibit potent strain-transcending neutralization. This suggests that RON2L alone cannot explain the production of strain-transcending antibodies. Additionally, a previous report by Srinivasan et al. in 2014 (<https://doi.org/10.1073/pnas.1409928111>) revealed that antibodies targeting RON2L, in the sera of rats immunized with RON2L alone, did not inhibit parasite growth in the growth inhibition assay (GIA). The study also demonstrated that inhibitory antibodies targeting AMA1 were crucial to parasite growth inhibition, constituting a large portion of IgG induced by AMA1-RON2L complex.

There is a second regrettable reason why we cannot perform depletion experiment requested. We currently do not have enough serum and total IgG available for depletion and GIA with multiple *P. falciparum* strains. We would have to initiate an additional animal study to obtain the required reagents for the proposed study. Never-the-less, we wish to reassure this reviewer that the experiment suggested, along with a number of others designed to examine the mechanism for strain-transcending protection, are a high priority for our group and will be evaluated in future studies.

Comment: Minor Comments

Line 159 - remove 'at'

Response: In line 159, 'at' has been removed in revised manuscript.

Comment: Line 615 – Was it polyethylene glycol or ethylene glycol that you used to cryoprotect?

Response: Polyethylene glycol (PEG) 400 was used to cryoprotect. We have corrected this in the revised manuscript.

Comment: Line 783 – In the BLI in Fig. 3 what was the top concentration of analyte used?

Response: Immunogens were two-fold serially diluted in HBS-EP+ buffer in the range of 200 nM to 3.125 nM. The concentration range was previously mentioned in the Methods section of the original manuscript. In response to this reviewer suggestion, we have now included the concentration range within the legend of Figure 3 of the revised manuscript.

Comment: Perhaps a supplementary figure with a sequence alignment of 3D7, FVO and Dd2 AMA1 sequences would be beneficial, as well as mapping polymorphisms onto the AMA1 structure with a particular focus on the loops?

Response: We thank the reviewer for identifying ways to improve the manuscript. In the revised manuscript, we have included a supplementary figure that illustrates a multiple

sequence alignment (MSA) of *P. falciparum* 3D7, FVO, and Dd2 AMA1 sequences along with the *P. falciparum* 3D7 AMA1 structure highlighting the polymorphisms observed among these three strains. We have provided this data in Supplementary Figure 6 and incorporated corresponding changes into the text. This revision does not change the inferences or conclusions presented in the original or revised manuscript.

Reviewers' Comments:

Reviewer #1:

Remarks to the Author:

Thank you to the authors for the letter of rebuttal. The amended manuscript satisfies all of the points that I had raised on initial review and in my opinion this manuscript is ready for publication.